# Tuning the interplay between nematicity and spin fluctuations in $Na_{1-x}Li_xFeAs$ superconductors

S.-H. Baek [1], Dilip Bhoi [2], Woohyun Nam[2], Bumsung Lee[2], D.V. Efremov[1], B. Büchner[1,3] & Kee Hoon Kim [2,4]

Strong interplay of spin and charge/orbital degrees of freedom is the fundamental characteristic of the iron-based superconductors (FeSCs), which leads to the emergence of a nematic state as a rule in the vicinity of the antiferromagnetic state. Despite intense debate for many years, however, whether nematicity is driven by spin or orbital fluctuations remains unsettled. Here, by use of transport, magnetization, and $^{75}$As nuclear magnetic resonance (NMR) measurements, we show a striking transformation of the relationship between nematicity and spin fluctuations (SFs) in $Na_{1-x}Li_xFeAs$; For $x \leq 0.02$, the nematic transition promotes SFs. In contrast, for $x \geq 0.03$, the system undergoes a non-magnetic phase transition at a temperature $T_O$ into a distinct nematic state that suppresses SFs. Such a drastic change of the spin fluctuation spectrum associated with nematicity by small doping is highly unusual, and provides insights into the origin and nature of nematicity in FeSCs.

[1] IFW Dresden, Helmholtzstr. 20, 01069 Dresden, Germany. [2] Department of Physics and Astronomy, Center for Novel State of Complex Materials Research, Seoul National University, Seoul 151-747, Korea. [3] Department of Physics, Technische Universität Dresden, 01062 Dresden, Germany. [4] Department of Physics and Astronomy, Institute of Applied Physics, Seoul National University, Seoul 151-747, Korea. Correspondence and requests for materials should be addressed to S.-H.B. (email: sbaek.fu@gmail.com) or to K.H.K. (email: optopia@snu.ac.kr)

Nematicity, i.e., spontaneous breaking of the $C_4$ symmetry of the crystal, has emerged as a research focus recently in the iron-based superconductors (FeSCs), because the nematic state can provide a clue to the understanding of high temperature superconductivity in these materials[1–8]. Currently two major scenarios have been proposed for the origin of nematicity: magnetic and charge/orbital[9]. The former assumes that the nematic state are entirely induced by the interband spin fluctuations (SFs). The latter scenario treats its origin as charge density waves or orbital orders.

The magnetic scenario is believed to be realized in the 122-family of FeSCs[10–12], particularly because a scaling relation was found between the spin fluctuations in nuclear magnetic resonance (NMR) and the shear modulus in the tetragonal phase of $Ba(Fe_{1−x}Co_x)_2As_2$[10]. On the other hand, the most simple compound FeSe is best described by the orbital scenario since nematic order occurs without any signature of the spin fluctuation enhancement[13–15]. In FeSCs other than FeSe, however, the SDW transition temperature $T_{SDW}$ is quite close to the nematic one $T_{nem}$, imposing limitations on investigating the interplay of nematicity and SFs in detail. Thus it is much desirable to find a system in which one can effectively tune SFs and nematicity in wide phase spaces, e.g., temperature and chemical doping.

From this point of view, NaFeAs, which is isostructural and isoelectronic to well investigated LiFeAs, is worth attention. LiFeAs shows only a superconducting (SC) ground state without a signature of nematicity or magnetism[16–18]. In contrast, NaFeAs is featured by the three successive transitions at low $T$; a nematic transition at $T_{nem}$~55 K is followed by a SDW at $T_{SDW}$~45 K and by a filamentary SC transition at ~8 K. In this respect, the study of (Na,Li)FeAs may allow a full spectrum of emergent orders coming from a delicate balance among competing orders by Li (Na)-substitution into Na (Li) layers in NaFeAs (LiFeAs). However, the study of such isoelectronic doping has been extremely challenging because (Na,Li)FeAs becomes easily phase separated due to distinct chemistry of Na and Li metals.

In this work, we report the successful growth of homogeneous $Na_{1−x}Li_xFeAs$ single crystals and the investigation of their electronic phase diagram up to $x = 0.1$. We found that with a systematic increase of $x$, the SDW is suppressed for $x \geq 0.03$, giving way to the SC state with the full Meissner shielding. Strikingly, also for $x \geq 0.03$, $^{75}As$ spin–lattice relaxation measurements show a sharp anomaly at a well-defined temperature $T_0$, evidencing a non-magnetic phase transition before entering the bulk SC state. Our comprehensive data further show that, above a critical doping $x_c$~0.03, spin and nematic fluctuations become strongly entangled, resulting in a charge/orbital ordered state below $T_0$. This implies that the nature of a nematic state could vary depending on the underlying electronic structure. Furthermore, our rich phase diagram strongly suggests that the normal state of FeSCs from which superconductivity emerges is far more complex than previously known.

## Results and discussion

### Crystal structure
Figure 1a presents the crystal structure and Fig. 1b shows the variation of the $c$-axis lattice parameter of the $Na_{1−x}Li_xFeAs$ single crystals, which decreases systematically with increasing $x$ up to 0.06 and then levels off from 0.08. The $c$ value was extracted from the (00l) reflections in the diffraction pattern measured along the $ab$-plane of the single crystals (Supplementary Fig. 1a) which suggests the absence of any other impurity phase. To determine the crystalline phase, we also performed the powder x-ray diffraction of the ground $Na_{0.95}Li_{0.05}FeAs$ single crystal (Supplementary Fig. 1b) and the pattern could be successfully refined by the tetragonal $P4/nmm$ structure as in the parent NaFeAs[19].

### Transport and magnetization measurements
The temperature ($T$) dependence of resistivity ($\rho$) of the selected $Na_{1−x}Li_xFeAs$ single crystals is displayed in the $T$ range from 3 to 300 K, and near the SC transition in Fig. 1d, e, respectively. Each resistivity curve was normalized by the value at 300 K ($\rho/\rho_{300K}$) and shifted vertically for clarity (for the original resistivity data, see Supplementary Fig. 2). For the undoped crystal, $\rho$ decreases smoothly exhibiting a typical metallic behavior with decreasing $T$, showing up several anomalous features at low $T$, an upturn at ~54 K (magenta arrow), a first drop (black arrow) at ~41 K, and a second drop to reach finally zero resistivity state (blue arrow) at ~7.7 K, which are identified as the nematic ($T_{nem}$), the SDW ($T_{SDW}$), and the SC ($T_c^\rho$) transition temperatures, respectively. $T_{nem}$ and $T_{SDW}$ appear more clearly in the derivative curves of $d\rho/dT$ as a deviation point (magenta arrow in Fig. 1f) and a local maximum or minimum (black arrow in Fig. 1f). At $T_c^\rho$ ~ 7.7 K, the corresponding magnetic susceptibility ($\chi$) (Fig. 1c) decreases abruptly, allowing us to assign $T_c$ from $\chi$, $T_c^\chi$ ~ 7.0 K. These transport and magnetization data are overall consistent with those found in previous reports on NaFeAs (refs.[19–24]).

With the same criteria applied on the parent NaFeAs, we could extract the $T_c^\rho$ ($T_c^\chi$), $T_{SDW}$, $T_{nem}$, and Meissner shielding fractions for the whole region ($0 \leq x \leq 0.1$). With increasing doping within $0 \leq x \leq 0.02$, both $T_{nem}$ and $T_{SDW}$ are rapidly suppressed. For $x = 0.03$, the $d\rho/dT$ curve reveals a jump which is assigned to $T_{nem}$, but does not show an anomaly associated with $T_{SDW}$. The magnetization data show that the SC volume fraction in the parent compound is very small in agreement with the previous results and increases weakly with doping up to $x$~0.03. Upon further doping ($x = 0.04$, 0.05, and 0.06), the SC shielding fraction at $T = 2$ K reaches 68%, 98%, and 90%, respectively, and the highest $T_c = 12.3$ K is obtained for $x = 0.05$, constituting an optimal doping. After the optimal doping, inserting more dopants into the system suppress both the SC shielding fraction and $T_c$, and eventually superconductivity disappears above $x = 0.12$.

### $^{75}As$ nuclear magnetic resonance
We now turn to $^{75}As$ NMR measurements on $Na_{1−x}Li_xFeAs$. The $^{75}As$ nuclei (nuclear spin $I = 3/2$) possess a large quadrupole moment. For axial symmetry, a magnetic field $H$ perpendicular to the principal axis (crystallographic $c$-axis in our case) yields two satellite lines whose separation is given by the quadrupole frequency $\nu_Q$. Figure 2a shows the $^{75}As$ NMR full spectrum at $H = 9.1$ T parallel to the $ab$-plane as a function of Li doping in the tetragonal/paramagnetic (PM) phase (at 60 K). Clearly, $\nu_Q = 9.93$ MHz for undoped crystal does not change notably with doping, indicating that the local symmetry or the average electric field gradient (EFG) at the $^{75}As$ is insensitive to Li dopants up to $x = 0.06$. On the other hand, the linewidth of the spectra progressively increases with increasing doping, which is naturally understood by the increase of chemical disorder due to dopants. It should be noted that the relative linewidth of the satellites with respect to the central one, which could be considered as a measure of chemical homogeneity, remains reasonably small up to ~10 at $x = 0.06$, in support of the high quality of our samples. Importantly, we do not observe other NMR lines with doping which would have indicated the presence of an impurity phase such as pure LiFeAs. Thus, the evolution of the $^{75}As$ spectra with doping provides local evidence for the successful incorporation of Li dopants into the Na layers of NaFeAs.

In the magnetically ordered state of NaFeAs, the stripe-like arrangement of Fe moments in the $ab$-plane, by symmetry,

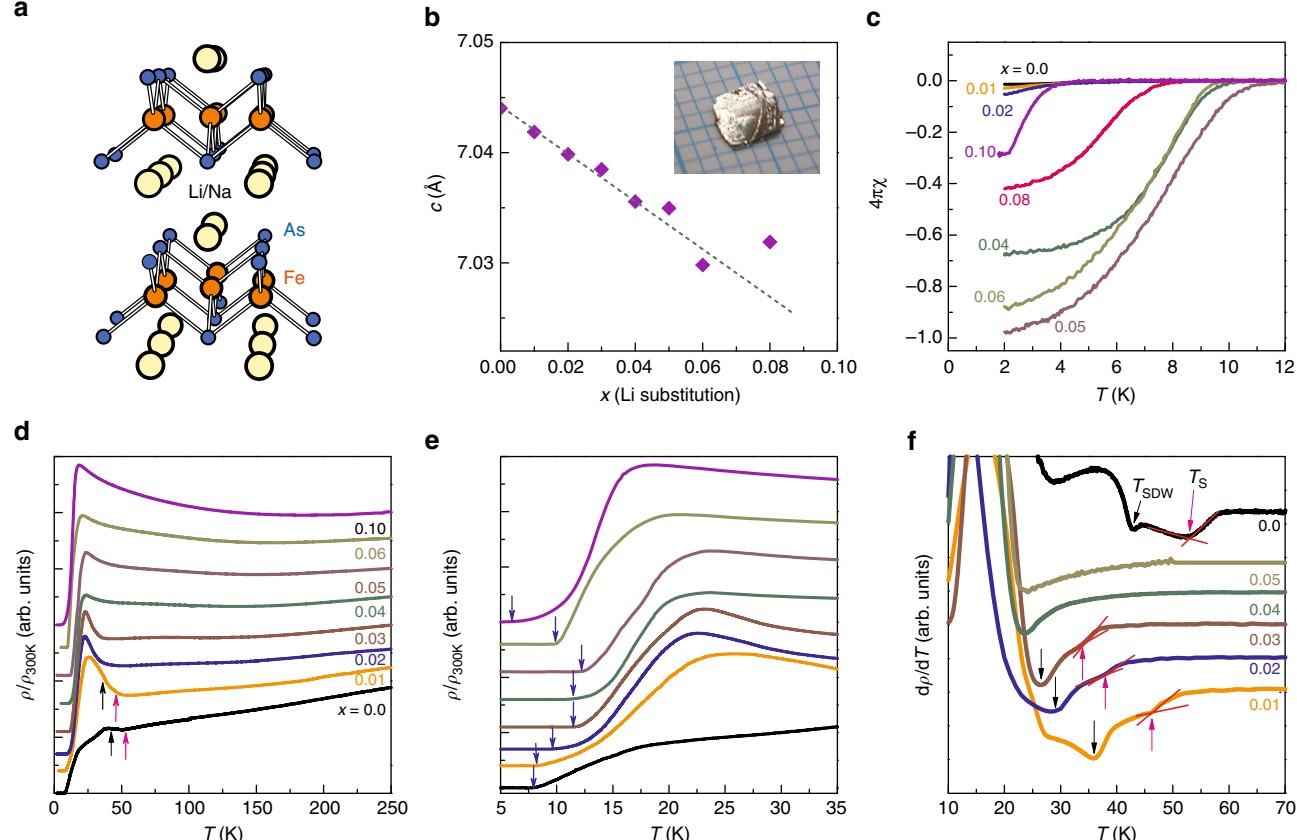

**Fig. 1** Characterization of the Na$_{1-x}$Li$_x$FeAs single crystals. **a** Crystal structure of Na$_{1-x}$Li$_x$FeAs. **b** Variation of the $c$-axis lattice parameter of the Na$_{1-x}$Li$_x$FeAs single crystals at 300 K with a linear dashed guide line. Inset shows the $ab$-plane image of a single crystal in a millimeter scale. **c** Zero-field cooled magnetizations at low temperatures at $H_{ab} = 10$ Oe, revealing SC transition and shielding fractions. **d–f** Temperature dependence of normalized resistivity, $\rho/\rho_{300K}$, enlarged resistivity near the SC transition, and d$\rho$/d$T$ for selected samples. Black and magenta arrows in **d**, **f** denote SDW ($T_{SDW}$) and structural ($T_S$) transitions, respectively, while blue arrows in **e** reflect the $T_c^\rho$ evolution with $x$. Note that although previous studies of NaFeAs have observed a local inflection point in the $\rho$ curve at $T_{SDW}$ (or a sharp dip in d$\rho$/d$T$), we observed weak minimum for our NaFeAs due to the presence of twinning

produces hyperfine fields at $^{75}$As pointing along the $c$-axis[25, 26]. As a direct consequence of two oppositely aligned antiferromagnetic (AFM) sublattices, the $^{75}$As central line splits into two AFM lines for $H \parallel c$. In case of $H \parallel ab$, the total magnetic field that $^{75}$As experiences slightly increases due to the vector sum of the local field along $c$ and the external field along $ab$, and the magnetic broadening of $^{75}$As line occurs accordingly. Therefore, long range AFM order in Na$_{1-x}$Li$_x$FeAs can be easily confirmed by NMR via a positive shift and a broadening of the $^{75}$As line for $H \parallel ab$, and a large AFM splitting of the line for $H \parallel c$. Indeed, for the parent ($x = 0$) and underdoped ($x = 0.02$) samples, the $^{75}$As line for $H \parallel ab$ broadens and shifts to higher frequency (Fig. 2b, c), and at the same time the $^{75}$As line for $H \parallel c$ splits into two well-defined AFM lines (Fig. 3f), thereby proving locally the SDW order. Note that the splitting of the $^{75}$As line shown in Fig. 2b is due to nemacitiy. Indeed, we determined $T_{nem}$ for $x = 0$ and 0.02 by measuring the $T$ dependence of $^{75}$As satellite line (Supplementary Fig. 3).

For $x \geq 0.04$, on the other hand, the $T$ dependence of the $^{75}$As spectrum is clearly told apart from those for $x \leq 0.02$ samples. First of all, there is no signature of a static SDW ordering. The $^{75}$As line preserves its shape without a shift nor a significant broadening down to low temperatures (see also Supplementary Fig. 4). Second, the $T$ evolution of the $^{75}$As spectrum is very similar for the two field orientations $H \parallel ab$ and $H \parallel c$, which contrasts sharply with the strongly anisotropic behavior observed

for $x \leq 0.02$. These features indicate that the system for $x \geq 0.04$ remains paramagnetic. Remarkably, the intermediate doping, i.e., $x = 0.03$, yields a very peculiar feature which seemingly separates the two doping regions, $x \leq 0.02$ and $x \geq 0.04$. That is, the $^{75}$As line is considerably broadened below $T_0 \sim 32$ K for both field orientations (see Figs. 2d and 3c.). The nearly isotropic NMR line broadening indicates that an inhomogeneous (short-ranged) magnetism develops. Moreover, we emphasize that $T_0$ for $x = 0.03$ is higher than $T_{SDW}$ for $x = 0.02$ and coincides with $T_{nem}$ (Fig. 4c). Therefore, we conclude that the inhomogeneous line broadening observed for $x = 0.03$ below $T_0$ is irrelevant to the SDW, but arises from an emerging phase which may involve a short-ranged magnetism.

For $x \geq 0.03$, alongside the suppression of the SDW, we observe that the resonance frequency $\nu$ of the $^{75}$As line in the PM phase is abruptly reduced. This behavior is clearly shown in Fig. 3g in terms of the Knight shift, $\mathcal{K} \equiv (\nu - \nu_0)/\nu_0 \times 100\%$ where $\nu_0$ is unshifted Larmor frequency. Since the second order quadrupole shift vanishes for $H \parallel c$, $\mathcal{K}_{H \parallel c}$ is equivalent to the local spin susceptibility $\chi_{spin} = \mu_B^2 N_F$ where $N_F$ is the density of states at the Fermi level. While a gradual reduction of $\chi_{spin}$ or $N_F$ with doping is commonly observed in FeSCs[27–29], the abrupt large reduction of $\chi_{spin}$ induced by a moderate doping is very unusual (Supplementary Fig. 5). This may suggest a modification of the Fermi surface geometry near $x \sim 0.03$, owing to, e.g., a Lifshitz transition[30].

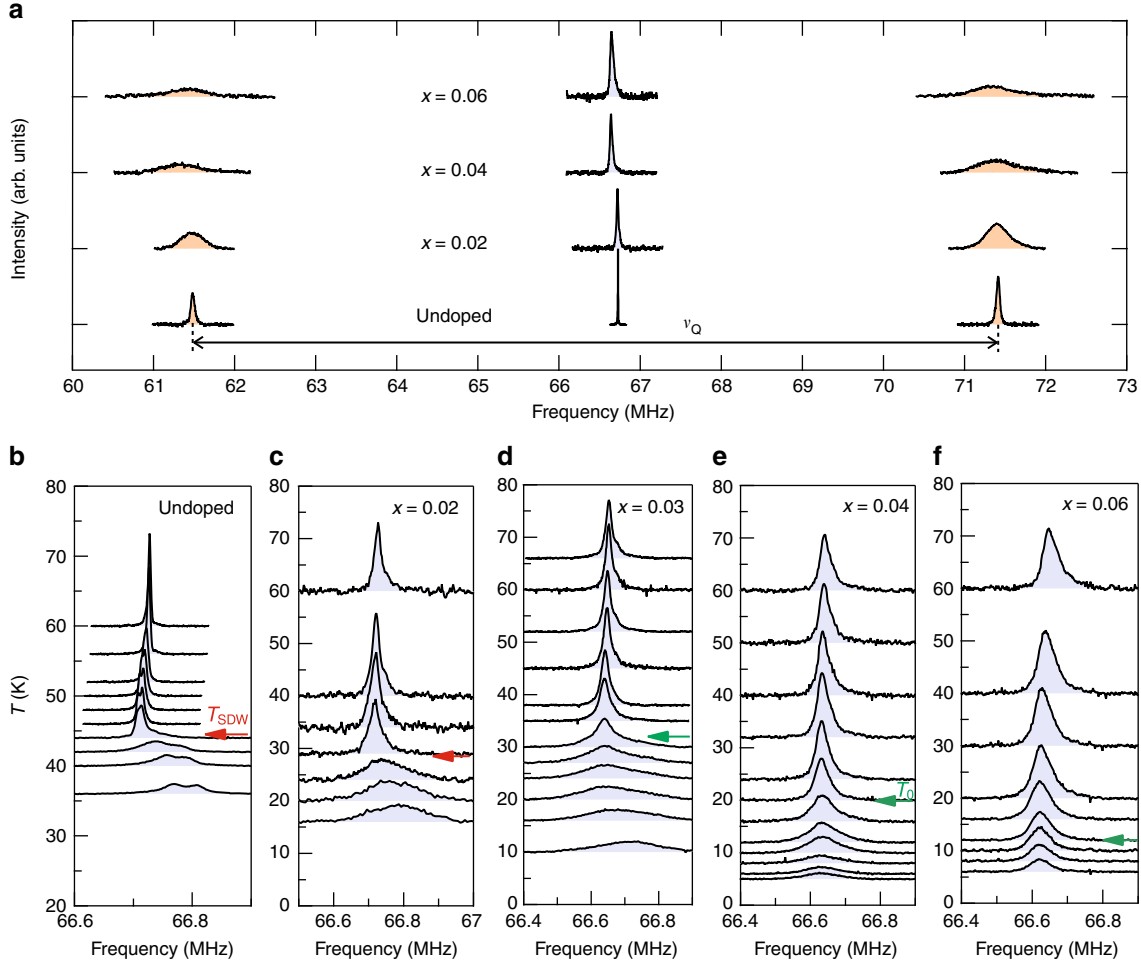

**Fig. 2** $^{75}$As NMR spectra in Na$_{1-x}$Li$_x$FeAs for for $H \parallel ab$. **a** Central and satellite lines of $^{75}$As as a function of doping at a fixed temperature of 60 K at 9.1 T. While the linewidth of the spectra increases with increasing doping, there is no other lines found and the quadrupole frequency $\nu_Q$ remains nearly unchanged. **b–f** Temperature dependence of the $^{75}$As central line as a function of $x$. For $x = 0$ and 0.02, the $^{75}$As spectrum starts to broaden and shift to higher frequency below $T_{SDW}$. In contrast, for $x \geq 0.04$, neither a significant broadening nor a shift of the line was observed, evidencing the absence of static magnetic moments. An inhomogeneous line broadening below $T_0$ for the intermediate doping $x = 0.03$ is ascribed to short-range magnetism associated with the charge/orbital ordered phase. The large reduction of the signal intensity below ~10 K is due to bulk superconductivity

**Low energy spin fluctuations.** Having established that static SDW order is abruptly suppressed at $x \sim 0.03$, we now investigate low energy spin dynamics, as probed by the spin–lattice relaxation rate divided by temperature $(T_1 T)^{-1}$, which is proportional to SFs at very low energy. $(T_1 T)^{-1}$ as a function of $T$ and $x$ are shown in Fig. 4. For $x = 0$ and 0.02, the diverging behavior of $(T_1 T)^{-1}$ is immediately followed by an exponential drop with decreasing $T$. The drastic change of $(T_1 T)^{-1}$ at $T_{SDW}$ precisely reflects two important characteristics of a SDW transition. The divergence of $(T_1 T)^{-1}$ at $T_{SDW}$ represents the critical slowing down of SFs toward long-range magnetic order and the subsequent exponential drop of $(T_1 T)^{-1}$ implies the depletion of low-energy spin excitations, i.e., the opening of a SDW gap. As doping is increased to 0.03, the divergent $(T_1 T)^{-1}$ observed for $x \leq 0.02$ is greatly suppressed, being consistent with the disappearance of static SDW order for $x \geq 0.03$, as discussed above. Unexpectedly, however, $(T_1 T)^{-1}$ drops rapidly below $T_0 > T_c$ forming a peak, indicating a phase transition at $T_0$. With further increasing doping, the $(T_1 T)^{-1}$ peak gradually moves to lower $T$, but its shape and height remain nearly the same.

The phase transition at $T_0$ observed for $x \geq 0.03$ cannot be of magnetic origin. First, the $T$ dependence of the $^{75}$As spectrum

(Figs. 2 and 3) does not show any signature of static magnetism, particularly for $x \geq 0.04$. Second, $(T_1 T)^{-1}$ or SFs does not diverge at $T_0$, implying that the magnetic correlation length remains short at $T_0$. Third, the sudden reduction of SFs between $x = 0.02$ and 0.03 is hardly observed in other FeSCs in which SFs above $T_{SDW}$ is gradually suppressed with increasing doping or pressure toward the optimal region[27, 31, 32]. Moreover, for $x = 0.03$, it turns out that $T_0$ nearly coincides with $T_{nem}$, as shown in Fig. 4c, implying that the $T_0$ transition is closely related to nematicity. Note that the sharp $(T_1 T)^{-1}$ peak at $T_0$ and its doping dependence are well distinguished from those arising from a glassy spin freezing observed in Co-doped BaFe$_2$As$_2$[33, 34]; either significant magnetic broadening or strong doping and field dependence of the $(T_1 T)^{-1}$ peak expected for a glassy magnetic state is not observed.

Interestingly, we find that the strong anisotropy of $(T_1 T)^{-1}$ above $T_{SDW}$ for $x \leq 0.02$ is maintained above $T_0$ for $x \geq 0.03$ (see Fig. 4a, b). That is, SFs for $H \parallel ab$ is factor of four stronger than for $H \parallel c$ over the whole doping range investigated. The robust spin fluctuation anisotropy with or without a static SDW order indicates that dynamic SDW fluctuations persist at least up to $x = 0.06$.

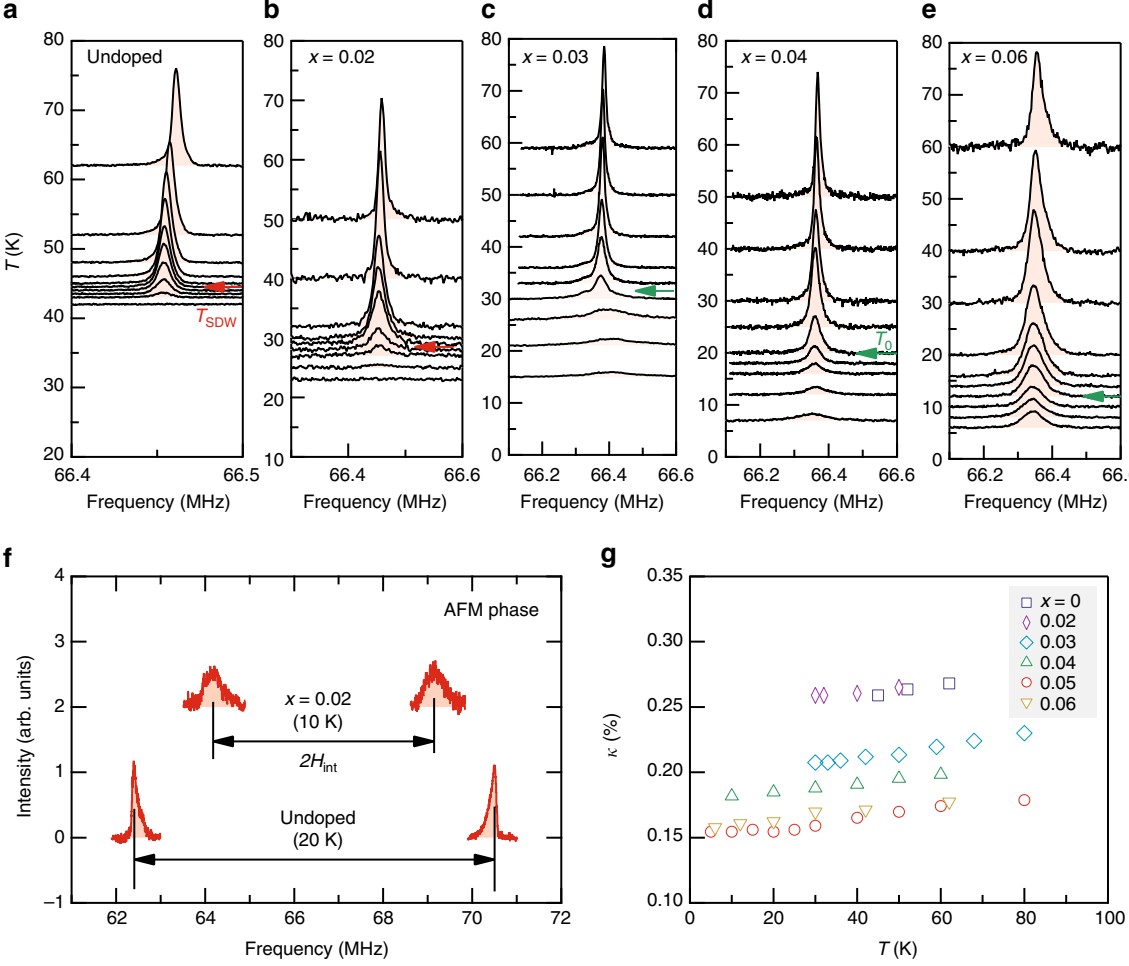

**Fig. 3** $^{75}$As NMR spectra in Na$_{1-x}$Li$_x$FeAs for $H \parallel c$. **a–e** Temperature dependence of the central line as a function of Li doping $x$ measured at 9.1 T. For $x = 0$ and 0.02, the $^{75}$As line rapidly disappears below $T_{SDW}$ due to the large hyperfine field resulting from the static Fe spin moment arranged antiferromagnetically. For $x \geq 0.03$, the $^{75}$As intensity remains finite down to low temperatures without a significant broadening, indicating the absence of long range magnetic order. **f** The AFM split $^{75}$As lines were detected at low temperatures for $x = 0$ and 0.02, manifesting the commensurate AFM phase. **g** The Knight shift $\mathcal{K}$ as a function of temperature and doping at 9.1 T parallel to $c$. The almost constant $\mathcal{K}$ for a given $x$ is rapidly reduced for $x \geq 0.03$, suggesting a change of the Fermi surface geometry

It may be worthwhile to note that below $T_0$ the $^{75}$As signal amplitude is notably reduced for both $H \parallel ab$ and $H \parallel c$, as shown in Figs. 2d–f and 3c–e (see also Supplementary Fig. 4). The suppression of signal intensity indicates that the volume fraction of the sample seen by NMR decreases in the charge/orbital ordered phase. This phenomenon is quite similar to the wipeout effect observed in the charge stripe phase of cuprate superconductors, in which NMR relaxation rates of the nuclei in spin-rich regions become too fast to be detected[35–37]. The underlying mechanism of the signal wipeout in Na$_{1-x}$Li$_x$FeAs remains unclear and needs further investigation. Interestingly, the similar wipeout of the NMR signal was also observed in the $^{77}$Se NMR study of FeSe in the nematic phase which does not involve any static magnetism[38].

**Phase diagram**. Figure 5 presents the temperature-doping phase diagram determined by our NMR and transport/magnetization measurements. When compared to the phase diagram previously known in NaFe$_{1-x}$A$_x$As ($A$ = Co, Cu, or Rh)[23, 24, 39, 40], a difference is the seeming mutual repulsion of the SDW and bulk superconductivity near $x_c \sim 0.03$, similar to that reported in LaFeAsO$_{1-x}$F$_x$[41] and NaFe$_{1-x}$Co$_x$As[42]. However, our data are not

sufficiently dense to conclude whether the SDW and SC phases coexist in the narrow doping range near $x = 0.03$ or completely repel each other.

The most important feature in Fig. 5 is the emergence of a non-magnetic phase below $T_0 > T_c$ at optimal doping. Whereas $T_0(x)$ is reasonably connected to $T_{nem}$ for $x \leq 0.02$, we note that the phase below $T_0$ cannot be a simple nematic because the strong suppression of SFs below $T_0$ (or spin–gap behavior) is unlikely due to nematicity itself. Although theory predicts that the nematic transition could trigger a pseudogap behavior[43], such a pseudogap is only viewed as magnetic precursors whose signature is a sharp increase of the magnetic correlation length. Therefore, we conclude that the phase below $T_0$ should involve a charge/orbital order which could give rise to a featured gap in the spin fluctuation spectrum.

An even more remarkable observation is the critical change of the relationship between SFs and nematicity with doping. For $x \leq 0.02$, SFs are enhanced just below the nematic transition at $T_{nem}$ and diverge at $T_{SDW}$. For $x \geq 0.03$, however, a strong enhancement of SFs precedes the phase transition into a charge/orbital nematic state at $T_0$, but it is suppressed once the charge/orbital nematic state develops. Theoretically, it has been proposed that

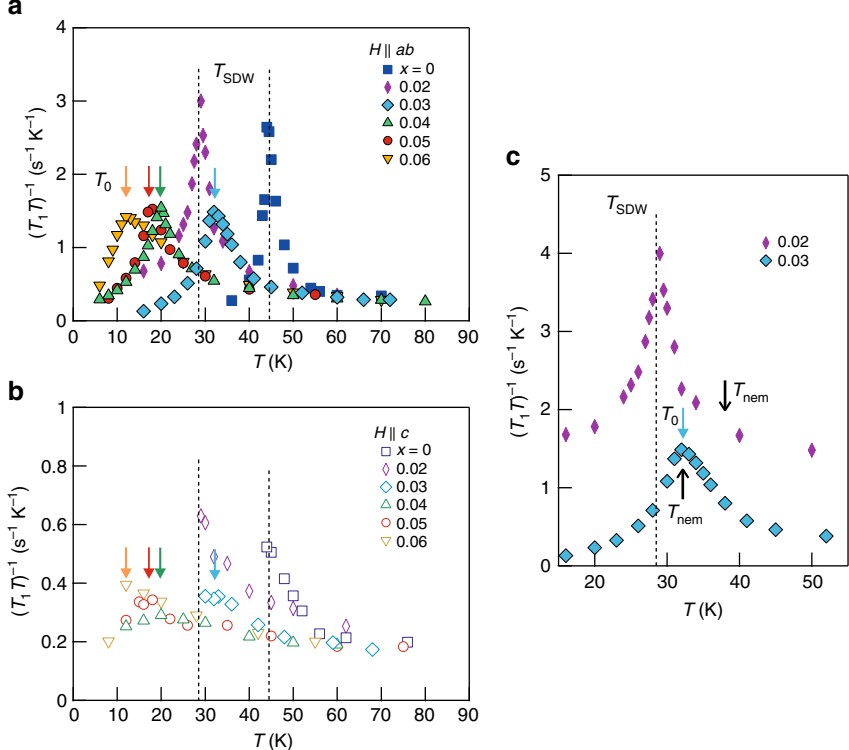

**Fig. 4** Spin–lattice relaxation rates as a function of temperature and doping. **a, b** The spin–lattice relaxation rate divided by temperature, $(T_1T)^{-1}$, measured at 9.1 T perpendicular and parallel to $c$, respectively. For $x = 0$ and 0.02, the SDW transitions were identified by the sharp peak of $(T_1T)^{-1}$ at $T_{SDW}$. Similar sharp transitions were observed for larger dopings ($x \geq 0.03$) at temperatures denoted by $T_0$, without an indication of long range SDW order. The comparison between **a** and **b** reveals that the strong anisotropy of spin fluctuations persists up to $x = 0.06$. **c** Comparison between the $(T_1T)^{-1}$ data for $x = 0.02$ and $x = 0.03$ obtained with $H \parallel ab$. The data for $x = 0.02$ are offset vertically for clarity. It shows that $T_0$ for $x = 0.03$ is higher than $T_{SDW}$ for $x = 0.02$, but nearly coincides with $T_{nem}$ obtained by resistivity (see Fig. 1)

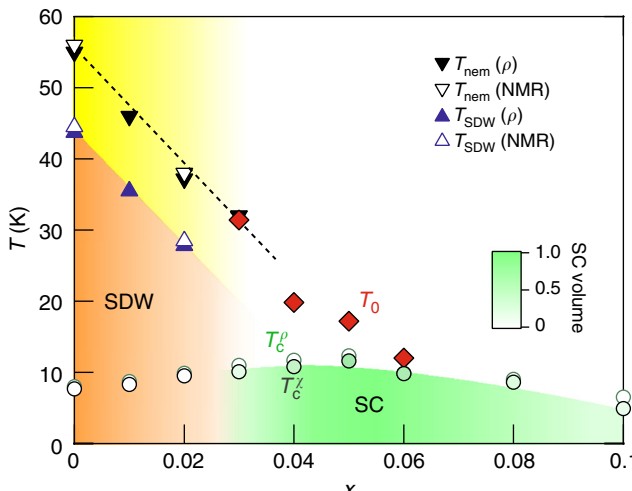

**Fig. 5** Temperature-doping phase diagram of Na$_{1-x}$Li$_x$FeAs. $T_{SDW}$, $T_{nem}$, $T_c^\rho$, and $T_c^\chi$ were obtained by our transport, magnetization, and NMR measurements which turn out to be consistent one another. The emerging phase at $T_0$ before entering the bulk SC state is seemingly connected to the nematic transition at lower dopings. The shade of green below $T_c$ schematically represents the SC volume fraction obtained from Fig. 1c

the strong interplay of spin and charge/orbital degrees of freedom could result in a charge density wave (CDW) state in proximity to a SDW state[9, 44–46]. Following the work[9], the competition of these two orders can be described by the effective Ginzburg–Landau functional:

$$\Delta F = \int_{\mathbf{q}} \left( \chi_s^{-1}(\mathbf{q})\left(\mathbf{M}_x^2 + \mathbf{M}_y^2\right) + \chi_c^{-1}(\mathbf{q})\left(\Phi_x^2 + \Phi_y^2\right) \right)$$
$$+ \frac{u}{2}\int_{\mathbf{q}}\left(\mathbf{M}_x^2 + \mathbf{M}_y^2 + \Phi_x^2 + \Phi_y^2\right)^2 \qquad (1)$$
$$- \frac{g}{2}\int_{\mathbf{q}}\left(\mathbf{M}_x^2 - \mathbf{M}_y^2 + \Phi_x^2 - \Phi_y^2\right)^2,$$

where the $\chi_s(\mathbf{q}, \Omega_n) \sim [\xi_s^{-2} + \alpha_s(q - Q_{x,y})^2]^{-1}$ and $\chi_c(\mathbf{q}, \Omega_n) \sim [\xi_c^{-2} + \alpha_c(q - Q_{x,y})^2]^{-1}$ are dynamical spin and charge susceptibilities, $\mathbf{M}_{x,y}$ and $\Phi_{x,y}$ are SDW and CDW order parameters with the propagation vectors $Q_x = (\pi, 0)$ and $Q_y = (0, \pi)$, respectively. The coupling constant $g$ in the leading order arises due to small non-ellipticity of the electron and hole pockets and is much smaller than $u$. The onset of nematic order parameter $\phi$ with $\mathbf{M}_{x,y} = \Phi_{x,y} = 0$ leads to renormalization of the magnetic correlation length $\xi_s^{-2} \to \xi_s^{-2} \pm \phi$ (see refs.[9, 43]). It leads to strong enhancement of SFs below $T_{nem}$. The opposite situation happens when the nematic transition coincides with the CDW phase $\Phi_x \neq 0$. The magnetic correlation length changes as $\xi_s^{-2} \to \xi_s^{-2} + (u \pm g)\Phi_x^2$ leading to suppression of SFs since $(u \pm g) > 0$. This may account for the drastic change of the relationship between nematicity and SFs. Our findings establish Na$_{1-x}$Li$_x$FeAs as a rich playground for the study of the interplay of spin and charge/orbital degrees of freedom in FeSCs.

## Methods

**Crystal growth and characterization.** High-quality single crystals of Na$_{1-x}$Li$_x$-FeAs were grown by a self-flux technique. Due to high reactivity of metallic Na, Li, Fe and As, all preparation processes were carried out inside an Ar-filled glovebox of

which $O_2$ and $H_2O$ contents were less than 1 ppm. Pure elemental Na (99.995%, Alfa Aesar), Li (99.9 + %, Sigma Aldrich), As (99.99999 + %, Alfa Aesar) lumps and Fe (99.998%, Puratonic) powder in a molar ratio (Na,Li):Fe:As = 3:2:4 were placed into an alumina crucible, then kept inside a welded Nb container under ~0.8 bar of Ar atmosphere. The welded container was finally sealed in an evacuated quartz ampoule. The ampoule was heated directly up to 1050 °C, stayed at this temperature for 1 h, afterward slowly cooled down to 750 °C with a rate of 3 °C/h and then heater was turned off while the ampoule was still kept in the furnace. 2D plate-like-shaped single crystals with shiny $ab$-plane surfaces were mechanically detached from a flux and typical sizes were around ~4⋅4⋅0.3 mm$^3$. Since the grown crystals were highly sensitive to air and moisture, all steps of physical measurement preparations were also done in an Ar-filled glovebox. The crystalline phase was determined by the powder x-ray diffraction using Cu $K_\alpha$ radiation at room temperature. To avoid oxidation and compensate the preferred orientation of single crystals, a sealed quartz capillary was adopted with grounded as-grown crystals for powder diffraction measurements. To measure the (00$l$) reflection peak, a piece of single crystal with the shiny $ab$-plane surfaces was sandwiched between kapton tapes. Rietveld refinements of the diffraction pattern were performed using Fullprof program. In particular, inductively coupled plasma atomic photoemission spectroscopy (ICP-AES) was carried out on a piece of optimally doped Na$_{0.95}$Li$_{0.05}$FeAs to check the elemental composition of Na and Li; a molar ratio of Na and Li was found as Na: Li = 0.944: 0.056, which is close to the expected composition. Electrical resistivity was measured by the conventional four probe technique using a conductive silver epoxy in PPMS™ (Quantum Design). Magnetic susceptibility was measured with a vibrating sample magnetometer in PPMS™ and MPMS™ (Quantum Design).

It should be noted that SC transitions of all the samples were considerably broad as the estimated transition width $\Delta T$ extracted at temperatures of 10 and 90% of resistivity maximum was 7.5–15 K. Moreover, all the resistivity curves exhibited a temperature window of showing insulating behavior (d$\rho$/d$T$ < 0) before the onset of SC transition similar to NaFe$_{1-x}$Cu$_x$As system[23]. In other doped Na 111 systems like NaFe$_{1-x}$Co$_x$As[22] and NaFe$_{1-x}$Rh$_x$As[24], robust metallic behavior were observed in broad doping ranges. The former Cu doping was claimed to have isoelectronic doping while the latter two involving large electron doping was inevitably accompanied by large chemical potential shift. As Li does not bring additional charge carriers to the systems, the effect of disorder-induced electron localization on transport is likely pronounced to result in the insulating-like behaviors as in the case of NaFe$_{1-x}$Cu$_x$As system.

**Nuclear magnetic resonance**. $^{75}$As (nuclear spin $I = 3/2$) NMR was carried out in Na$_{1-x}$Li$_x$FeAs single crystals at an external field of 9.1 $T$ and in the range of temperature 4.2–100 K. Due to the extreme sensitivity of the samples to air and moisture, all the samples were carefully sealed into quartz tubes filled with Ar gas. The sealed sample was reoriented using a goniometer for the accurate alignment along the external field. The $^{75}$As NMR spectra were acquired by a standard spin-echo technique with a typical $\pi/2$ pulse length 2–3 μs. The nuclear spin–lattice relaxation rate $T_1^{-1}$ was obtained by fitting the recovery of the nuclear magnetization $M(t)$ after a saturating pulse to following fitting function,

$$1 - M(t)/M(\infty) = A[0.9\exp(-6t/T_1) + 0.1\exp(-t/T_1)]$$

where $A$ is a fitting parameter. We also measured $^{23}$Na NMR spectra for $x = 0.04$. As shown in Supplementary Fig. 6, we confirm that the $^{23}$Na is barely influenced by the $T_0$ phase transition.

**Data availability**. The data that support the findings of this study are available from the corresponding authors (S.H.B. or K.H.K.).

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

## Acknowledgements

This work was financially supported by the National Creative Research Initiative (2010-0018300) and Global Collaborative Research Projects (2016K1A4A3914691) through Korea's NRF, which is funded by Ministry of Science, ICT and Future Planning (MSIP). The work at Germany has been supported by the Deutsche Forschungsgemeinschaft (Germany) via DFG Research Grants BA 4927/2-1. D.V.E. acknowledges VW-foundation for partial financial support.

## Author contributions

K.H.K. and B.S.L. have proposed and initiated the project. W.H.N., B.S.L., D.B. have grown single crystals and characterized transport and structure properties. S.H.B. performed NMR measurements and analyzed data; S.H.B., D.V.E., and K.H.K. participated in writing of the manuscript. All authors discussed the results and commented on the manuscript.

## Additional information

**Competing interests:** The authors declare no competing interests.

