## [Peer Review File · Nature Communications]

Reviewers' comments:

Reviewer #1 (Remarks to the Author):

The paper by Baek et al reported synthesis of iron-based superconductor $(\text{Na}_{1-x}\text{Li}_x)\text{FeCo}$ and characterizations by transport, magnetization and nuclear magnetic resonance measurements. The main conclusion is that, above a critical doping $x \sim 0.03$, a charge density wave (CDW) like nematic state appears.

Nematic order is an interesting phenomenon in iron-based superconductors. The authors intended to address this issue through synthesizing a mixed compound between NaFeAs and LiFeAs . The motivation is well justified. However, there are many concerns about the sample quality and the claims.

(a) About the sample quality. The authors found a first-order like phase boundary between SDW and SC and claim that it is common in FeSCs. As is well-known now, the early claim of phase separation between SDW and SC was due to poor sample quality. After sample quality is improved, it has been demonstrated firmly that the two orders coexist at a microscopic length scale. See, for example, Laplace, et al, Phys. Rev. B 80, 140501(R) (2009); Li et al, Phys. Rev. B 86, 180501(R) (2012). Such fact should be described and the authors should discuss if or not they can rule out a phase separation around $x = 0.03 \sim 0.04$.

(b) The authors found a peak at T_0 in the temperature dependence of $1/T_1T$ and conclude to a CDW like charge/orbital order. There are serious questions about this claim. Firstly, it is questionable that a CDW like charge/orbital order will give rise to a peak, see for example Imai and Lee, arXiv:1712.09720v1. I have not seen any example in which CDW results in a peak of $1/T_1T$. Secondly, in the nematic state, T_{1a} and T_{1b} should be different as shown by Zhou et al, Phys. Rev. B 93, 060502 (2016). In this work, the authors seem to have only measured T_1 with the field applied perpendicular to the plane. I suggest the authors to consider this issue by performing further detailed measurements.

(c) The resistivity data should be shown, rather than normalized resistance.

In summary, this manuscript lacks both proper reference to previous works that are closely related to the current one and important for understanding the results presented, and convincing discussions. It is unable to judge the validity of the claims from the present writing.

Reviewer #2 (Remarks to the Author):

This paper is a very well-written and presented account of an experimental investigation into Li-doped NaFeAs . The authors have made some excellent crystal samples and have shown that the NMR linewidth increases with Li doping, demonstrating that the Li mixes in well. They have determined the phase diagram and have linked their results with the way in which spin fluctuations are first promoted and then suppressed as the nematic phase is altered. I am impressed with the detailed work that lies behind these studies. The references are all appropriate. I think this work presents important science and very much deserves to be published. Therefore, I recommend that this interesting paper is accepted for Nature Communications.

I have a few small comments:

* The authors say that the reduction of the spin susceptibility, as deduced from the Knight shift, is "unprecedented" and may be related to band structure effects (and possibly a Lifshitz transition). Is there any indication from the known band structures of NaFeAs and LiFeAs that this might be

the case?

* The authors have done structural analysis of their samples. They should state how the lattice parameter varies with Li doping.

* On page 8, a sentence begins without a definite article: "Quite opposite situation happens...." I suggest this is changed to "The opposite situation happens...."

Reviewer #3 (Remarks to the Author):

It has proved difficult to synthesize homogeneous samples of the iron pnictide $\text{Na}_{1-x}\text{Li}_x\text{FeAs}$ system. The authors argue that they have succeeded in doing so and they establish its phase diagram from bulk magnetization, resistivity and NMR experiments. They argue that as x is increased there is a first order transition towards a new nematic-CDW state.

The data is of high quality and the topic is interesting because electron nematicity in these materials is not well understood. The interpretation proposed by the authors is plausible but there are too many concerns with the sample characterization and with the interpretation to recommend publication of the paper in its present form.

Below is a list of the main problems:

Page 3: referencing is a bit deficient. It is uncommon to attribute nematicity in the pnictides to charge density waves (the traditional presentation is spin fluctuations vs. spontaneous orbital order) so references and explanations would help. At least a review paper should be cited. I think the success of the spin-fluctuation scenario does not rely only on ref. 7 but on a whole array of experimental works, including neutron scattering data for example. Again, at least a review paper should be cited. When mentioning the absence of spin fluctuation enhancement in FeSe, please cite Bohmer et al. PRL 2015.

One would expect the present results to be discussed, even briefly, in the context of previous NMR studies of the same or similar materials. Perhaps surprisingly, such studies are not cited (in page 7 when mentioning the NaFeCoAs phase diagram or elsewhere). For example, Phys. Rev. B 84, 054528 (2011), PRB 88, 134518 (2013), PRB 90, 144502 (2014), Phys. Rev. B 93, 060502(R) (2016). Aren't these works relevant?

Page 3: How can we be sure that there are no nematic fluctuations in LiFeAs ? Is the high temperature NMR broadening recently reported in different NMR studies of Fe based SC absent in this compound?

The transport data for $x=0$ look different from those in PRB 85, 224521 or PRB 91, 020508. At T_N , a dip was observed in $d\rho/dT$, not a bump. So, one wonders whether the interpretation of resistivity data is unambiguous and whether SDW can really be excluded by this data (especially in small portions of the sample).

Page 5: isn't it surprising that the local charge environment around 75As is insensitive to Li dopants? Doesn't this suggest that there is some phase separation, with the signal from one of the phases not being observed?

Page 5: when talking about linewidth no numbers are given (throughout the paper). We need to see a plot of the linewidth as a function of doping. "Reasonably small" is not a sufficiently precise information.

Given the intensity wipeout from 75As NMR, it is important that the authors provide ^{23}Na NMR

data. This could also be useful to test the possibility of phase separation on some length scale or more generally inhomogeneity.

It is also important that the authors provide a plot of the NMR signal intensity and of the linewidth versus temperature for the different samples. They are speaking of "moderate line broadening below T_0 " which is ill defined and the corresponding data is not shown. Could the decrease of $1/T_1T$ at low temperature be an artifact of a wipeout for the doped samples?

Page 6: there is a significant jump in the Knight shift values from $x=0.02$ to $x=0.03$. It would be helpful that the authors compare the amplitude of this jump to literature data in order to convince us that this cannot be due to a doping difference. From refs 18 and 19, I can see a trend but I don't know how much the doping effectively changes with Co. Again, giving numbers is important in order to convince the reader that there is no inhomogeneity issue with the doped samples.

As far as I understood it, the wipeout effect in cuprates has been attributed to slow spin fluctuations, which are actually triggered by charge order. So it is only indirectly related to CDW order. Therefore, the observation of wipeout cannot be taken as evidence of CDW like state. Actually, it rather suggests slow spin fluctuations, which would contradict the authors' scenario.

Throughout the paper, the non-magnetic state is declared to include the 0.03 concentration. However, there is a clear shift of the line (Fig. 2d) just as for $x=0$ and $x=0.02$. Therefore, the situation does not seem to be clear for this sample. How do the authors explain this shift?

On page 8, the authors rely on theory predicting an imaginary CDW. Yet, in the rest of the paper they simply mention CDW. One would like to know what is the difference between CDW and iCDW and maybe have a more precise idea of the proposed electronic state.

What we really learn from this work, why it is important enough to be published in Nature Commun. is a little unclear. The sentences ending the abstract and the introduction are not totally convincing in this respect.

Hereby, we resubmit our manuscript entitled “Tuning the interplay between nematicity and spin fluctuations in $\text{Na}_{1-x}\text{Li}_x\text{FeAs}$ superconductors”, which was reviewed by three experts. We would like to appreciate all reviewers for their valuable time and effort. We are certain that their thoughtful comments and suggestions are really helpful to improve and strengthen our manuscript.

Reviewer 2 was very positive and gave a few minor comments which were answered and reflected in the revised manuscript.

Reviewers 1 and 3 raised a couple of questions and concerns, some of which are overlapped. We have considered seriously all of their comments in this revised manuscript and the Supplementary Information (SI). The summary of changes in this revision is as follows (the revised text in the manuscript were coloured in blue for convenience).

- (1) Following comments made by Reviewers 1 and 3, we added more references which are closely related to the current one in the revised manuscript.
- (2) We revised paragraphs in the abstract and in the introduction to emphasize the significance of our results.
- (3) Many phrases and paragraphs in the manuscript were revised in order to reflect concerns and suggestions made by reviewers (marked by blue color).
- (4) We newly prepared Supplementary Information to include further experimental details.
- (5) For detailed responses to the comments of reviewers, please see below.

We believe that we fully addressed all the criticism and concerns given by reviewers in this reply and the revised manuscript.

Sincerely,

Seung-Ho Baek

On behalf of the authors

Reviewers' comments:

Reviewer #1 (Remarks to the Author):

The paper by Baek et al reported synthesis of iron-based superconductor $(\text{Na}_{1-x}\text{Li}_x)\text{FeCo}$ and characterizations by transport, magnetization and nuclear magnetic resonance measurements. The main conclusion is that, above a critical doping $x \sim 0.03$, a charge density wave (CDW) like nematic state appears.

Nematic order is an interesting phenomenon in iron-based superconductors. The authors intended to address this issue through synthesizing a mixed compound between NaFeAs and LiFeAs . The motivation is well justified. However, there are many concerns about the sample quality and the claims.

(a) About the sample quality. The authors found a first-order like phase boundary between SDW and SC and claim that it is common in FeSCs. As is well-known now, the early claim of phase separation between SDW and SC was due to poor sample quality. After sample quality is improved, it has been demonstrated firmly that the two orders coexist at a microscopic length scale. See, for example, Laplace, et al, Phys. Rev. B 80, 140501(R) (2009); Li et al, Phys. Rev. B 86, 180501(R) (2012). Such fact should be described and the authors should discuss if or not they can rule out a phase separation around $x=0.03 \sim 0.04$.

Reply: We understand the reviewer's concern about the sample quality. Clearly, as pointed by the reviewer, the coexisting SC and SDW phases were verified to be intrinsic, particularly, in the Ba122 materials. In general, however, we believe that the coexistence of the two phases is determined by the properties of a system, e.g., three dimensionality, the ellipticity of the electron bands, etc, rather than by the sample quality. [See, for example, Vavilov et al, Supercond. Sci. Technol. 23, 054011 (2010)]

It is well known that NMR is one of the most sensitive tool to sample quality, because the NMR spectrum is strongly influenced by sample inhomogeneity, particularly for a quadrupole nucleus such as ^{75}As . As we have already described in detail in the text, our ^{75}As NMR spectrum gradually increases with increasing doping as expected, but still remains very narrow less than 50 kHz even for $x=0.06$. (see the attached figure below) Furthermore, our NMR data clearly show that the doped system remains a single phase (see Fig. 2a and relevant discussion). Therefore, these strongly support that our single crystals are of high quality.

Thus far, the sharp phase boundary has been reported in F-doped LaFeAsO system [Luetkens et al, Nat. Mat. (2009)] and in Co-doped NaFeAs [Ma et al, PRB (2014)], but our phase diagram suggests that Li-doped NaFeAs may also be the case. Based on this observation and by noting that Ba122 materials are much more 3D-like than LaFeAsO and NaFeAs, we raised the possibility that the otherwise competing SDW and SC orders in 2D systems could be turned into the coexistence as the system's dimensionality is getting close to 3D, as in Ba122.

Nevertheless, since we cannot rule out the possibility that SDW and superconductivity coexist in a very small doping range near $x=0.03$ and this issue is not a main point in this manuscript, we carefully rewrote the relevant paragraphs to avoid any confusion or possible controversy.

(b) The authors found a peak at T_0 in the temperature dependence of $1/T_1T$ and conclude to a CDW like charge/orbital order. There are serious questions about this claim. Firstly, it is questionable that a CDW like charge/orbital order will give rise to a peak, see for example Imai and Lee, arXiv:1712.09720v1. I have not seen any example in which CDW results in a peak of $1/T_1T$.

Reply: First of all, we would like to point out that NMR cannot detect charge order directly in general, because NMR is basically the spin probe. But NMR may serve as indirect probe for charge ordering, depending on how it affects the spin fluctuation spectrum. In La-based cuprates, for example, the charge stripe order triggers the formation of spin-rich regions in which spin fluctuations are slowed down, which in turn causes either the strong enhancement of $1/T_1$ of ^{139}La or the wipeout of the ^{63}Cu signal. Here, the different behavior of the two nuclei is likely due to the different strength of the hyperfine coupling [see our relevant papers, Baek et al, PRB 92, 155144 (2015) & Baek et al, PRB 96, 094519 (2017)].

In our case, an important characteristic which allows NMR to detect the phase transition into charge/orbital order is a gap behavior in the spin excitation spectrum, which is responsible for the sharp drop of $1/T_1T$ below T_0 . Since there is no signature of long-range magnetism at optimal dopings, we attributed the phase transition at T_0 which involves a spin gap behavior to ordering in the charge/orbital channels.

Please note that the upturn of $1/T_1T$ above T_0 is not related to charge/orbital order, but to antiferromagnetic instability which persists even for $x > 0.03$. (See our discussion in the text regarding the persistence of AFM spin fluctuations in the whole doping range investigated.) Therefore, the peak of $1/T_1T$ is the consequence of the sudden suppression of the AFM spin fluctuations owing to the spin gap feature associated with charge/orbital ordering.

In this revision, we removed the word “CDW-like” because it may give a wrong impression that the charge/orbital order in our study is similar to the charge stripe order in cuprates.

Secondly, in the nematic state, T_{1a} and T_{1b} should be different as shown by Zhou et al, Phys. Rev. B 93, 060502 (2016). In this work, the authors seem to have only measured T_1 with the field applied perpendicular to the plane. I suggest the authors to consider this issue by performing further detailed measurements.

Reply: We agree that T_{1a} and T_{1b} in the nematic state should be distinguishable, as it was shown in the NMR study of FeSe done by some of the authors [Baek et al, Nature Mater, 14, 210 (2015)].

In fact, we were able to detect the clear splitting of a ^{75}As satellite line in $\text{Na}_{1-x}\text{Li}_x\text{FeAs}$ for $x=0$ and $x=0.02$, which allowed us to determine the nematic transition temperature.

However, the splitting and consequently the anisotropy of T_{1a} and T_{1b} was not resolved anymore for $x > 0.03$ due to the line broadening larger than the splitting. We attached below the plot of the satellite spectrum, which shows that the splitting is not observed for $x=0.04$. (This figure was included in the Supplementary Information (SI).)

While our phase diagram strongly suggests that $T_0(x)$ is connected to nematicity as both critical temperatures are in the same line in the phase diagram, we emphasize in the manuscript that the charge/orbital ordered phase for $x > 0.03$ is not a spin nematic anymore, as it is different in physical properties. For revealing the nature of the phase, one needs additional investigation beyond NMR.

(c) The resistivity data should be shown, rather than normalized resistance.

Reply: As suggested by the reviewer, we have given an additional figure in the SI (Fig. S2) showing the temperature dependence of the resistivity of $\text{Na}_{1-x}\text{Li}_x\text{FeAs}$ single crystals with different Li substitution. To clearly display the evolution of different transition temperature with doping, we have shown the normalized resistance in the main text. For the reviewer, we have attached the figure below.

In summary, this manuscript lacks both proper reference to previous works that are closely related to the current one and important for understanding the results presented, and convincing discussions. It is unable to judge the validity of the claims from the present writing.

Reply: We thank the reviewer for the useful comments. As Reviewer 3 also pointed out missing references, we paid more attention in this revised manuscript in citing previous works which are closely related to our work, as much as we could.

For other comments, we believe that we fully addressed the concerns raised by the reviewer and that our manuscript was greatly strengthened accordingly.

Reviewer #2 (Remarks to the Author):

This paper is a very well-written and presented account of an experimental investigation into Li-doped NaFeAs. The authors have made some excellent crystal samples and have shown that the NMR linewidth increases with Li doping, demonstrating that the Li mixes in well. They have determined the phase diagram and have linked their results with the way in which spin fluctuations are first promoted and then suppressed as the nematic phase is altered. I am impressed with the detailed work that lies behind these studies. The references are all appropriate. I think this work presents important science and very much deserves to be published. Therefore, I recommend that this interesting paper is accepted for Nature Communications.

Reply: We appreciate the reviewer for careful reading and the positive comments.

I have a few small comments:

* The authors say that the reduction of the spin susceptibility, as deduced from the Knight shift, is "unprecedented" and may be related to band structure effects (and possibly a Lifshitz transition). Is there any indication from the known band structures of NaFeAs and LiFeAs that this might be the case?

Reply : Our collaborators are currently working on $\text{Na}_{1-x}\text{Li}_x\text{FeAs}$ by ARPES.

* The authors have done structural analysis of their samples. They should state how the lattice parameter varies with Li doping.

Reply : Due to the high sensitivity of the sample to oxidation, it was difficult for us to perform the powder x-ray diffraction analysis for every sample so far. Hence, we provided only powdered XRD for selected doping as shown in the Fig.1 of main text. Instead, we could manage to measure the (00l) peaks in the most of the single crystal series as attached below (we also provided it in the SI). The presence of only c-axis (00l) reflections suggest the absence of any other impurity phase. Based on the data, we extracted the c-axis parameter with Li substitution, which shows systematic decrease with the increase of Li substitution up to 0.06 and then levels off from 0.08.

* On page 8, a sentence begins without a definite article: "Quite opposite situation happens...." I suggest this is changed to "The opposite situation happens...."

Reply : Following the suggestion, we corrected the sentence.

Reviewer #3 (Remarks to the Author):

It has proved difficult to synthesize homogeneous samples of the iron pnictide $\text{Na}_{1-x}\text{Li}_x\text{FeAs}$ system. The authors argue that they have succeeded in doing so and they establish its phase diagram from bulk magnetization, resistivity and NMR experiments. They argue that as x is increased there is a first order transition towards a new nematic-CDW state.

The data is of high quality and the topic is interesting because electron nematicity in these materials is not well understood. The interpretation proposed by the authors is plausible but there are too many concerns with the sample characterization and with the interpretation to recommend publication of the paper in its present form.

Below is a list of the main problems:

Page 3: referencing is a bit deficient. It is uncommon to attribute nematicity in the pnictides to charge density waves (the traditional presentation is spin fluctuations vs. spontaneous orbital order) so references and explanations would help. At least a review paper should be cited. I think the success of the spin-fluctuation scenario does not rely only on ref. 7 but on a whole array of experimental works, including neutron scattering data for example. Again, at least a review paper should be cited. When mentioning the absence of spin fluctuation enhancement in FeSe, please cite Bohmer et al. PRL 2015.

One would expect the present results to be discussed, even briefly, in the context of previous NMR studies of the same or similar materials. Perhaps surprisingly, such studies are not cited (in page 7 when mentioning the NaFeCoAs phase diagram or elsewhere). For example, Phys. Rev. B 84, 054528 (2011), PRB 88, 134518 (2013), PRB 90, 144502 (2014), Phys. Rev. B 93, 060502(R) (2016). Aren't these works relevant?

Reply: Following the reviewer's suggestion, we added three relevant review papers : Bohmer et al, C. R. Physique (2015), Yi et al, Quantum Materials (2017), Bohmer et al, J. Phys: condens. matter (2018). And we added the reference, Bohmer et al, PRL (2015) and other relevant references mentioned by the reviewer. In addition, we also made efforts to find and cite missing references throughout the whole text.

Page 3:How can we be sure that there are no nematic fluctuations in LiFeAs ? Is the high temperature NMR broadening recently reported in different NMR studies of Fe based SC absent in this compound?

Reply: We realized that our previous writing was somewhat inaccurate because the nematic susceptibility in LiFeAs has not been accurately determined to our knowledge. So we modified the phrase “without a trace of nematic fluctuations” in the text to “without a signature of nematicity”.

The transport data for $x=0$ look different from those in PRB 85, 224521 or PRB 91, 020508. At T_N , a dip was observed in $d\rho/dT$, not a bump. So, one wonders whether the interpretation of resistivity data is unambiguous and whether SDW can really be excluded by this data (especially in small portions of the sample).

Reply : This is a very careful point of the reviewer, which we have also noticed already. Indeed, to confirm the quality of the Li undoped sample (i.e. NaFeAs), we have spent long time to change the synthesis conditions. For example, we have grown the NaFeAs single crystal using different flux ratio, of which transport data are summarized below in the left panel. On the right panel, we have plotted the derivative of the corresponding data. Irrespective of the different flux ratio used, we can clearly see two anomalies in the derivative curve, which is located at 43 K and 53 K, respectively, in consistent with T_{SDW} and T_s as determined from the NMR measurements and with other references (Deng et al., PRB, 91, 020508 (2015)). Therefore, our data clearly support that regardless of the detailed shape of $d\rho/dT$ curve, the SDW and the nematic (structural) transition become robust to appear at almost same transition temperatures.

One important clue to understand the various transport shape (different $d\rho/dT$ curve shape) for the undoped sample has come up from a recent reference as also pointed out by the reviewer. Indeed, as proven in the reference (Deng et al., PRB, 91, 020508 (2015)), the shape of the resistivity, ρ_{ab} and related derivative curve is highly sensitive to the twinning present in the crystals. The anomaly in ρ_{ab} , at the SDW and structural transitions is not so sharp for the twinned crystals, being qualitatively similar to our data.

Whereas, for the de-twinned crystals, the anomaly become sharper for resistance, R_b , as measured along b-axis and as a result, dR_b/dT curve exhibits a very sharp dip at the T_{SDW} . Therefore, our conclusion is that different shape of dp/dT curve which doesn't show clearly a dip, is due to different amount twins present in the sample rather than due to other effects such as impurities.

In the original version, we agree that we put the arrow indicating T_N in a wrong position (near the bump) in Fig. 1(f) and made a confusion. To be more consistent with the published data, we now assign the dip position of dp/dT curve as T_{SDW} . This will provide a consistent assignment for T_{SDW} with the literature and relieve the concern on the quality of the samples particularly on low doping side. Below is the temperature dependence of the normalized in-plane resistance of the twinned and de-twinned NaFeAs single crystals taken from Ref. (Deng et al., PRB, 91, 020508 (2015)).

Page 5: isn't it surprising that the local charge environment around ^{75}As is insensitive to Li dopants? Doesn't this suggest that there is some phase separation, with the signal from one of the phases not being observed?

Reply : We realized that our previous writing was misleading. Indeed, there is clear change of the local charge environment around ^{75}As with doping — the NMR line broadening which increases with increasing Li doping (as demonstrated in Fig. 2a) means that the local charge environment becomes inhomogeneous with doping, which is precisely what is anticipated in the presence of dopants. What we meant was that the “average” electric field gradient is insensitive to Li doping, which is not surprising because the atomic structure is usually insensitive to small dopings. We thank the reviewer for pointing out this mistake, which was corrected as follows.

“Clearly, $\nu_Q=9.93$ MHz for undoped crystal does not change notably with doping, indicating that the local symmetry or the average electric field gradient (EFG) at the ^{75}As is insensitive to Li dopants.”

Page5: when talking about linewidth no numbers are given (throughout the paper). We need to see a plot of the linewidth as a function of doping. “Reasonably small” is not a sufficiently precise information.

Reply: We agree with this criticism, and we plotted FWHM vs T as a function of doping (see below). The figure clearly shows that for $x > 0.03$ the FWHM is weakly broadened below T_0 , compared to the clear magnetic broadening for $x \leq 0.03$. We added this figure in the SI.

Given the intensity wipeout from ^{75}As NMR, it is important that the authors provide ^{23}Na NMR data. This could also be useful to test the possibility of phase separation on some length scale or more generally inhomogeneity.

Reply: We measured and compared ^{23}Na spectra for $x=0.04$ above and below T_0 , as shown below. One could see that ^{23}Na spectra are not notably influenced by the T_0 transition. This reflects that the ^{75}As in the FeAs plane effectively probes the Fe sites, excluding the possible phase separation.

It is also important that the authors provide a plot of the NMR signal intensity and of the linewidth versus temperature for the different samples. They are speaking of “moderate line broadening below T_0 ” which is ill defined and the corresponding data is not shown. Could the decrease of $1/T_1T$ at low temperature be an artifact of a wipeout for the doped samples?

Reply: We thank for the useful suggestion, and made the plot of the (normalized) NMR intensity vs temperature as shown below. (We added this figure in the SI.) The plot reveals that the wipeout takes place only for $x > 0.03$, while the intensity is preserved down to low temperatures within experimental error for $x \leq 0.03$.

We emphasize that the sharp decrease $1/T_1T$ below T_0 cannot be due to the wipeout or any other artifacts. This is because any kind of additional contribution to $1/T_1$ should be added, i.e., $1/T_1$ is always additive. Furthermore, the wipeout effect is ascribed to the slowing down of spin fluctuations in some region of the sample and thus it would have enhanced $1/T_1T$. Therefore the gap behavior should arise from the intrinsic phase transition of the system.

Page 6: there is a significant jump in the Knight shift values from $x=0.02$ to $x=0.03$. It would be helpful that the authors compare the amplitude of this jump to literature data in order to convince us that this cannot be due to a doping difference. From refs 18 and 19, I can see a trend but I don't know how much the doping effectively changes with Co. Again, giving numbers is important in order to convince the reader that there is no inhomogeneity issue with the doped samples.

Reply : We thank the reviewer for the suggestion. Below we directly compared the ^{75}As Knight shift data in Co-doped BaFe_2As_2 (Ning et al) taken at 150 K and those in Li-doped NaFeAs taken at 50 K. Indeed, we think that this comparison will help readers convinced of the rapid drop of the Knight shift near $x \sim 0.03$. We also added this figure in the SI.

As far as I understood it, the wipeout effect in cuprates has been attributed to slow spin fluctuations, which are actually triggered by charge order. So it is only indirectly related to CDW order. Therefore, the observation of wipeout cannot be taken as evidence of CDW like state. Actually, it rather suggests slow spin fluctuations, which would contradict the authors' scenario.

Reply: The reviewer is correct that the wipeout in cuprates is due to the slowing down of SFs in the spin-dominant regions, which is triggered by charge order (see also our response to the comment by Reviewer 1).

Before going into the detailed response, we would like to point out a common confusion among researchers: the wipeout of the ^{63}Cu signal in cuprates means that ^{63}Cu NMR does not "see" the signal from the spin-rich regions. In other words, the slow spin fluctuations is not detected by ^{63}Cu NMR!! On the other hand, the ^{139}La signal remains strong in the stripe phase due to much weaker hyperfine coupling of ^{139}La than the ^{63}Cu . As a result, $1/T_1$ of ^{139}La can probe the slow spin fluctuations and reveals diverging behavior toward spin order, whereas that of ^{63}Cu is insensitive to charge order — See our recent ^{63}Cu and ^{139}La NMR study in $\text{La}_{2-x}\text{Sr}_x\text{CuO}_4$ which demonstrates the contrasting behavior of the two nuclei [Baek et al, PRB 96, 094519 (2017)].

Namely, although the wipeout of the ^{63}Cu NMR signal arises from slow SFs, the slow SFs does not affect $1/T_1T$ of the remaining signal. A similar line of

reasoning can be used in our case, and thus the wipeout does not contradict the suppression of $1/T_1T$.

It is interesting to note that the similar wipeout effect was also observed by ^{77}Se NMR in the nematic phase of FeSe. [See the discussion in Imai et al, PRL 102, 177005 (2009)] Therefore, we may argue that the wipeout is, either directly or indirectly, related to charge/orbital order or nematicity.

Regardless, the origin of the signal wipeout below T_0 remains unclear at the moment. To avoid an unnecessary confusion, we revised the relevant paragraphs significantly and moved them to the middle of the text.

Throughout the paper, the non-magnetic state is declared to include the 0.03 concentration. However, there is a clear shift of the line (Fig. 2d) just as for $x=0$ and $x=0.02$. Therefore, the situation does not seem to be clear for this sample. How do the authors explain this shift?

Reply: As the reviewer mentioned, the ^{75}As line for $x=0.03$ broadens and slightly shifts below T_0 , which partly resembles the data below T_N for $x=0$ and 0.02 . In stark contrast, however, the temperature dependence of $1/T_1T$ for $x=0.03$ is clearly distinguished from the data for $x=0$ and 0.02 , but is nearly identical with those observed for $x>0.03$. Based on our observations, we discussed in the text that magnetism observed for $x=0.03$ must be a short-ranged one, i.e., the correlation length remains short. Regardless of the short-range magnetism, the nuclei still experience the distributed local fields which can be aligned along the external field. (Note that in this case the direction of the local fields is not fixed along the c axis unlike for $x=0$ and 0.02 . This explains why we observe the similar temperature evolution of the spectrum for $H \parallel c$.) Thus a slightly larger effective field at the nuclei in an external field accounts for the small positive shift of the line.

On page 8, the authors rely on theory predicting an imaginary CDW. Yet, in the rest of the paper they simply mention CDW. One would like to know what is the difference between CDW and iCDW and maybe have a more precise idea of the proposed electronic state.

Reply: Although an imaginary CDW (i.e., a charge-current density wave) was specified in the original theoretical description, the GL functional [Eq. (1)] is a quite general form, and so it is not restricted to iCDW, but also applicable to other different form of charge/orbital order (e.g., stripe CDW, a bond charge ordering, etc) which competes with SDW and generates a spin gap behavior.

Without the loss of generality, we removed “imaginary” in the text to avoid a confusion in this revision. Also we decided not to use “CDW-like” in this

revised manuscript because any charge/orbital order could be viewed as a different form of CDW.

What we really learn from this work, why it is important enough to be published in Nature Commun. is a little unclear. The sentences ending the abstract and the introduction are not totally convincing in this respect.

Reply: In this work, we have shown that for $x > 0.03$ emerges a new phase which is distinguished from any known phases in FeSCs so far—The phase is featured by the simultaneous occurrence of a nematic and a novel charge/orbital orders just above the bulk superconducting state. A striking observation is that this phase *suppresses* SFs. It contrasts sharply with the known behavior in the nematic phase of the parent NaFeAs as well as of FeSe, in which the nematic transition *promotes* SFs.

Such a drastic transformation of the interplay between nematicity and spin fluctuations with small dopings has never been observed before in FeSCs, and thus our finding is an important step forward to fully understand nematicity in FeSCs.

We revised the last paragraph of the abstract and the introduction to emphasize the significance of our findings more clearly, as indicated by blue color. We hope that the reviewer is now convinced that our work deserves the publication in Nature Communications.

Reviewers' comments:

Reviewer #1 (Remarks to the Author):

The authors have replied to most of the questions/criticisms raised by the reviewers and greatly improved their manuscript. However, some parts of the newly added text are problematic or lack accuracy. I list two of them.

1) On page 8, the authors discuss about the phase diagram and claim that SDW and superconductivity (SC) are mutual repulsive. They took the old data of $\text{LaFeAsO}_{1-x}\text{F}_x$ as supportive evidence. As I commented in the first report, although early works suggested that SDW and SC are phase separated, subsequent experiments on improved samples indicated that it is not the case. This likely applies to $\text{LaFeAsO}_{1-x}\text{F}_x$ as well, as demonstrated by a recent work (arXiv:1707.04085). Therefore, the pertinent statement about SDW and SC should be modified. The issue is more related to sample quality rather than dimensionality or Fermi surface.

2) I believe that LiFeAs was discovered by Jin's group and shown by his collaborators to be non-magnetic. Therefore, in addition to Ref.16, the papers by his group [Solid State Commun. 148 (2008) 538.; J. Phys. Soc. Jpn. 79, 083702 (2010).] should be added.

Reviewer #3 (Remarks to the Author):

I have read the new version of the manuscript and fully acknowledge the authors' efforts to improve the manuscript and to answer the reviewers' questions. Yet, I am still not convinced by their interpretation of the data and by the possible impact of this work. It is a solid study with puzzling and interesting results but I do not see in the manuscript what "provides fresh insights into the origin and nature of nematicity in FeSCs", as the authors (over) state in the introduction.

The really puzzling observation is that their maximum in $1/T_1$ occurs at T_{nem} for the 0.03 sample. However, I fail to be convinced that this is not a coincidence. This sample is really atypical, right at junction between two different phases in the phase diagram. The maximum temperature in $1/T_1$ at higher doping is equally consistent with an extrapolation of T_{SDW} rather than of T_{nem} .

I disagree with the authors' discussion of the wipeout effect. I think their data rather suggest that long-range SDW order at $x < 0.03$ transforms into short-range glassy magnetic state at $x > 0.03$. This produces a BPP-type peak as in the cuprates and a signal wipeout because of too short T_2 due to slow spin fluctuations. The situation could well be analogous to (unfortunately uncited here) observations in 122 pnictides: A.P. Dioguardi et al. Phys. Rev. Lett. 111, 207201 (2013) and Phys. Rev. B 92, 165116 (2015).

I would advise the authors to be more open to alternative explanations and to consider publishing their work in Phys. Rev. B.

Hereby, we resubmit our manuscript entitled “Tuning the interplay between nematicity and spin fluctuations in $\text{Na}_{1-x}\text{Li}_x\text{FeAs}$ superconductors”. We would like to thank reviewers for reading and commenting on our work again. They acknowledged that our responses to the reviewers concerns were nicely done, and raised some further concerns.

We find that Reviewer #1 is not satisfied with the discussion about the sharp boundary between SDW and SC phases in the phase diagram. As we mentioned in our previous reply, we cannot rule out the possible coexistence of the two phases near $x=0.03$ and therefore we do not strongly object to the Reviewer 1’s concern. In this revision, we did not raise the possibility of the role of dimensionality and leave the interpretation open to readers (the modified text is marked by blue color). Also we added two references mentioned by him/her.

Reviewer #3 fully acknowledged our efforts to improve the manuscript and to answer the reviewer’s questions. Furthermore, he/she admits that our work is “a solid study with puzzling and interesting results”. Nevertheless, he/she is not convinced of our findings and interpretations, raising an alternative explanation that the peak of $1/T_1T$ at T_0 may be associated with a glassy magnetism. As responded below in detail, his suggestion is undoubtedly ruled out and our interpretation is solid. For readers, however, we added a sentence to state that the $1/T_1T$ peak at T_0 is unrelated to glassy magnetism observed in Co-doped Ba122 in the revised manuscript.

We believe that we addressed all the criticism and concerns given by reviewers in this reply and the revised manuscript.

Sincerely,

Seung-Ho Baek

On behalf of the authors

Reviewers' comments:

Reviewer #1 (Remarks to the Author):

The authors have replied to most of the questions/criticisms raised by the reviewers and greatly improved their manuscript. However, some parts of the newly added text are problematic or lack accuracy. I list two of them.

1) On page 8, the authors discuss about the phase diagram and claim that SDW and superconductivity (SC) are mutual repulsive. They took the old data of $\text{LaFeAsO}_{1-x}\text{F}_x$ as supportive evidence. As I commented in the first report, although early works suggested that SDW and SC are phase separated, subsequent experiments on improved samples indicated that it is not the case. This likely applies to $\text{LaFeAsO}_{1-x}\text{F}_x$ as well, as demonstrated by a recent work (arXiv:1707.04085). Therefore, the pertinent statement about SDW and SC should be modified. The issue is more related to sample quality rather than dimensionality or Fermi surface.

This issue is actually not important in our manuscript, and we reflected the reviewer's concern in the revision, removing the discussion of the possible role of dimensionality. Now we do not put an emphasis on the phase repulsion, but also open to the possible phase coexistence near $x=0.03$.

2) I believe that LiFeAs was discovered by Jin's group and shown by his collaborators to be non-magnetic. Therefore, in addition to Ref.16, the papers by his group [Solid State Commun. 148 (2008) 538.; J. Phys. Soc. Jpn. 79, 083702 (2010).] should be added.

We added the two references in the revised manuscript.

Reviewer #3 (Remarks to the Author):

I have read the new version of the manuscript and fully acknowledge the authors' efforts to improve the manuscript and to answer the reviewers' questions. Yet, I am still not convinced by their interpretation of the data and by the possible impact of this work. It is a solid study with puzzling and interesting results but I do not see in the manuscript what "provides fresh insights into the origin and nature of nematicity in FeSCs", as the authors (over) state in the introduction.

The really puzzling observation is that their maximum in $1/T_1$ occurs at T_{nem} for the 0.03 sample. However, I fail to be convinced that this is not a coincidence. This sample is really atypical, right at junction between two different phases in the phase diagram. The maximum temperature in $1/T_1$ at higher doping is equally consistent with an extrapolation of T_{SDW} rather than of T_{nem} .

We already described in detail in pages 6-7 of our manuscript that the phase transition at T_0 cannot be of magnetic origin, taking into consideration the doping and temperature dependences of ^{75}As NMR spectra and $1/T_1T$ at the same time. It appears that our arguments in the text were overlooked by the reviewer.

We agree that the data at $x=0.03$ is somewhat atypical, but it may be understandable, simply because it is located at the boundary between the two phases. Even so, the strong suppression of spin fluctuations as x increases from 0.02 to 0.03 means the strong suppression of the SDW instability at $x=0.03$. Therefore, in any case, T_{SDW} , if exist, cannot increase for $x=0.03$ compared to that for $x=0.02$. It is thermodynamically impossible. This means that T_0 for $x=0.03$, which is larger than T_{SDW} for $x=0.02$, cannot be the SDW transition temperature. We also discussed this already in the text, directly comparing the data for $x=0.02$ and 0.03 in Fig. 4c. Further, as revealed in Fig. 4a, the height and shape of the $1/T_1T$ peak for $x=0.03$ is essentially identical with those at higher dopings. Then, $T_0 \sim T_{nem}$ for $x=0.03$ is quite naturally connected to T_0 for $x>0.03$.

Accordingly, the reviewer's comment, "The maximum temperature in $1/T_1$ at higher doping is equally consistent with an extrapolation of T_{SDW} rather than of T_{nem} ." is hardly justified. *We did not extrapolate the points in the phase diagram at will without a physical ground.* In contrast, we connected T_0 and T_{nem} by combining the comprehensive experimental data set and the phase diagram together.

I disagree with the authors' discussion of the wipeout effect. I think their data rather suggest that long-range SDW order at $x<0.03$ transforms into short-range glassy magnetic state at $x>0.03$. This produces a BPP-type peak as in the cuprates and a signal wipeout because of too short T_2 due to slow spin fluctuations. The situation could well be analogous to (unfortunately uncited here) observations in 122 pnictides: A.P. Dioguardi et al. Phys. Rev. Lett. 111, 207201 (2013) and Phys. Rev. B 92, 165116 (2015).

We respect the reviewer's opinion of the wipeout effect, but we disagree with the view that the wipeout effect is related to a glassy magnetic state. (We are aware

that Nick Curro at UC Davis, the main author of the two references mentioned by the reviewer, has this view.) We would like to point out that there is no firm consensus of the exact origin of the wipeout effect even in cuprates. Therefore, the reviewer's opinion (and of course ours too!) regarding the wipeout effect is inevitably subjective at the moment. Regardless of the origin of the signal wipeout, however, our data and their analysis show that the $1/T_1T$ peak is totally irrelevant with glassy or any kind of magnetism as follows.

We already carefully considered all the possibilities before reaching our conclusions, including glassy spin freezing as an origin of the sharp peak of $1/T_1T$. As a matter of fact, our descriptions that the T_0 transition cannot be of magnetic origin (see pages 6-7 of the manuscript) are naturally applied to a glassy magnetic state as well. Most importantly, a significant NMR line broadening is unavoidable feature of a glassy spin frozen state. (In a glassy magnetic state, the spins are frozen at random directions, and therefore the local fields at nuclei at an external field are widely distributed, leading to a considerable NMR line broadening.) If the long-range SDW order transforms into a glassy state for $x > 0.03$, as the reviewer suggested, the NMR line below T_0 should be even broader than that observed in the long-range SDW state at $x=0$ and 0.02 . Obviously, this is not the case. As shown in Figs. 2 and 3, the ^{75}As line does not broaden notably below T_0 . In particular, the linewidth for $x=0.06$ does not even change at all below T_0 , despite the similar sharp peak of $1/T_1T$ (see also SI). It is impossible to explain this feature with a glassy magnetic state.

Furthermore, the height and shape of the $1/T_1T$ peak at T_0 remains intact with further increasing doping, while T_0 is progressively suppressed, which is incompatible with the expected strong doping dependence. (One can easily check the difference in the reference mentioned by the reviewer, Dioguardi, PRB 92, 165116 (2015)) We also verified that $1/T_1$ near T_0 has no field dependence, which is another unequivocal proof that the peak of $1/T_1T$ represents a true phase transition.

At this moment, we do not know precisely the ground state below T_0 , which calls for other studies beyond ours, but robust is the conclusion that the transition at T_0 cannot be of magnetic origin and thus should be ordering in the charge/orbital channel.

I would advise the authors to be more open to alternative explanations and to consider publishing their work in Pays. Rev. B.

We appreciated previous thoughtful and useful comments of the reviewer which led to a significant improvement of our manuscript, but we do not understand and cannot accept this advice.

We will be happy to consider any reasonable alternative explanations, but our conclusion that the T_0 transition represents ordering in the charge/orbital channel, is only one explanation that we could physically reason best within the constraints set by the experimental observations.